# Assessment of Human Health Risk Indices Due to Metal Contamination in the Surface Water of the Negro River Sub-Basin, Áncash

**DOI:** 10.3390/ijerph21060733

**Published:** 2024-06-05

**Authors:** Walter Bravo-Zevallos, Yadira Fernández-Jerí, Juan C. Torres-Lázaro, Karol Zuñiga-Bardales

**Affiliations:** 1Grupo de Investigación Bioquímica Toxicológica (BIOTOX), Facultad de Farmacia y Bioquímica, Universidad Nacional Mayor de San Marcos, Lima 15001, Peru; yfernandezj@unmsm.edu.pe; 2Instituto Nacional de Investigación en Glaciares y Ecosistemas de Montaña (INAIGEM), Huaraz 02002, Peru; jctorres@inaigem.gob.pe; 3Instituto Geológico, Minero y Metalúrgico (INGEMMET), Lima 15034, Peru; kzuniga@ingemmet.gob.pe

**Keywords:** metals, Negro River sub-basin, human health risk, acid rock drainage, pollution assessment, source apportionment

## Abstract

The accelerated loss of glacial cover in the Cordillera Blanca in Áncash, Peru, exposes the underlying rocks with high concentrations of sulfides from the Chicama Formation to oxidation and leaching processes, generating acid rock drainage (ARD) in glacial and periglacial areas. These are transported by surface runoff, contaminating the surface water with high concentrations of metals and sulfates, as well as increasing the acidity, which poses a risk to human health and the ecosystem. Therefore, the risk indices for human health due to metal contamination were evaluated at 19 surface water sampling points distributed in the Río Negro sub-basin. Hydrochemical analyses revealed average metal concentrations in the following order: Fe (28.597 mg/L), Al (3.832 mg/L), Mn (1.085 mg/L), Zn (0.234 mg/L), Ni (0.085 mg/L), Co (0.053 mg/L), Li (0.036 mg/L), Cu (0.005 mg/L), and Pb (0.002 mg/L). The risk was determined by calculating the Heavy Metal Pollution Index (HPI) and the Hazard Index (HI). The average HPI value was 360.959, indicating a high level of contamination (HPI ≥ 150). The human health risk assessment indicated that adverse effects caused by iron, lithium, and cobalt in children and adults should be considered. Through the use of Pearson correlation analysis, principal component analysis, and cluster analysis, it was identified that SO_4_^2−^, Fe, S, Al, Co, Mn, Ni, Zn, and Li originate from natural sources, associated with the generation of ARD in glacial and periglacial areas.

## 1. Introduction

Global warming is characterized by the increase in the average global temperature, caused by the accumulation of greenhouse gases (GHGs) and the evident influence of human activities on this process [1]. Between 2011 and 2020, there was a 1.1 °C increase in global temperature compared to the period from 1850 to 1900 [2]. The main GHGs are CO_2_ (75%), CH_4_ (16%), and N_2_O (2%), with their emissions largely due to the burning of fossil fuels [3]. The effects of climate change include the loss of glacial coverage, sea level rise, and an increase in the frequency and intensity of extreme weather events such as floods, storms, hurricanes, heatwaves, the loss of biodiversity, and deglaciation, among others [4,5,6,7,8,9]; these events impact human health, including the increase in zoonotic and vector-borne diseases, malnutrition and food insecurity, deaths related to natural disasters, and impact on mental health [10,11,12].

Peru presents seven of the nine vulnerability characteristics recognized by the United Nations Framework Convention on Climate Change (UNFCCC) [13], mainly due to its diverse geography, vulnerable population, dependence on natural resources, and the presence of glaciers.

Approximately 99% of the world’s tropical glaciers are located in the Andes Mountains, with 71% in Peru, 20% in Bolivia, 4% in Ecuador, and 4% in Colombia and Venezuela [14]. In Peru, there are 18 glacial mountain ranges covering 1114.11 km^2^ as of 2018, with the largest being the Cordillera Blanca (448.81 km^2^), Vilcanota (255.44 km^2^), and Vilcabamba (101.00 km^2^). However, climate change is contributing to the disappearance of glaciers, with a loss of 1284.95 km^2^ (53.56%) of glacial coverage from 1962 to 2018. Specifically, the Cordillera Blanca lost 274.56 km^2^ (37.95%) of its glacial coverage [15].

The loss of glacial coverage in the Cordillera Blanca exposes the underlying sulfide-rich rock associated with the Chicama Formation to oxidation and leaching processes, generating acidic rock drainage (ARD) in glacier and periglacial zones. These ARDs are characterized by high concentrations of metals, sulfates, and a distinctly acidic pH, generally below 4. These are transported by surface runoff, thus contaminating surface water resources [16].

Research conducted in the sub-basin of the Buín River and the Quillcay sub-basin, consisting of the Sallap, Cojup, and Quillcayhuanca micro-basins, provides concrete evidence of this phenomenon in the Cordillera Blanca [16,17,18,19,20]. Magnússon et al. [16] observed that the presence of waters with a low pH and high concentrations of aluminum, sulfate, and iron was associated with the loss of glacial coverage, especially in areas adjacent to the Chicama Formation. On the other hand, Martel et al. [19] identified that in the Shallap and Quillcayhuanca micro-basins, where the Chicama Formation predominates, the water resource has a pH lower than 4, accompanied by high concentrations of Fe, Al, Mn, Zn, Ni, and sulfates. Santofimia et al. [17], in the Pachacoto micro-basin, observed that springs, streams, and lakes affected by acidic rock deposition showed high concentrations of metals such as iron, aluminum, manganese, zinc, nickel, and cobalt, as well as high electrical conductivity, indicating intense mineralization.

In the Río Negro sub-basin, the retreat of glacial coverage from the snow-capped Cashán, Rurec, Uruashraju, Yanamarey, Pumahuacanca, and Tuctu mountains, associated with the Chicama Formation, increases the formation of acidic rock deposits (ARDs), rich in metals such as iron, aluminum, manganese, zinc, nickel, cobalt, and lithium, in addition to sulfates and with an acidic pH. This situation has severely contaminated the surface waters of the Tarahua, Huansan, Huamash lagoons, as well as the Rurec, Pumahuaganga, and Uquian streams, and the Negro and Olleros rivers [16,21,22,23,24,25,26,27,28].

This contamination of water quality poses a severe risk to the health of the communities in the district of Olleros and Recuay in the Áncash region. These populations rely exclusively on these water sources for both human consumption and essential economic activities, such as agriculture and livestock. Therefore, identifying and understanding these risks are crucial for designing mitigation strategies and implementing projects aimed at ensuring water resource quality and protecting the health of the exposed population.

The objective of this study was to evaluate the risk indices for human health due to metal contamination in the surface water of the Rio Negro sub-basin. The results of this study will allow for identifying which metals pose a risk to human health and their spatial distribution; moreover, it will be possible to identify the exposed populations in the area. This information will be fundamental for the implementation of decontamination projects in order to properly utilize the water resource and monitor the health of the exposed inhabitants.

## 2. Materials and Methods

### 2.1. Study Area

The Rio Negro sub-basin (9°38′40″ S; 77°23′12″ W) is located in the Olleros district, Ancash region, Peru. It covers an approximate area of 179.955 km^2^. The elevations range from 3300 to 5752 masl [29]. The district’s climate is characterized by two seasons: the flood season (from October to April) and the dry season (from May to September), with an average annual precipitation of 54.88 mm in 2022. The temperature ranges between 25 to −3.4 °C, with an average of 11.06 °C in 2022 [30,31]. The main river is the Olleros River, with a length of 6.319 km. This river flows into the Santa River, which is part of the Santa River basin, located on the Pacific watershed. The surface water in the area primarily originates from glacier melt (Cashán, Rurec, Uruashraju, Yanamarey, Pumahuacanca, Tuctu, and Pucaraju) and rainfall, which accumulates in lagoons (Tarahua, Huansan, and Collotacocha) and flows through gullies and rivers (Olleros and Negro) [23,24,25,26,32].

There are no mining or hydrocarbon environmental liabilities in the sub-basin [33,34]. The hydraulic infrastructure in the area consists of 11 intakes, 72 lateral channels, and 5 diversion channels [35]. The study area includes a portion of Huascarán National Park and encompasses two urban populated centers (Huaripampa and Olleros) and 30 rural populated centers with 2946 inhabitants [36,37,38]. Nineteen sampling points were selected, as shown in Figure 1 and Appendix A.

### 2.2. Metal Analysis

#### 2.2.1. Sample Collection and Processing

Sampling and in situ determination of parameters (pH, dissolved oxygen (DO), electrical conductivity (EC), total dissolved solids (TDS), salinity, and temperature) at 19 points located in the Rio Negro sub-basin were carried out during the dry season (August) in 2022. Water sampling was simple, and the choice of sampling points was non-probabilistic for convenience. At each location, surface water was collected using high-density polyethylene (HDPE) plastic bottles of 250 mL, at the midpoint of the stream, avoiding shallow areas [39]. Additionally, field and transport blanks were prepared with deionized water to assess possible cross-contamination during sampling execution, transport, or due to contaminated preservatives.

Nineteen samples for metal analysis were acidified with concentrated nitric acid (HNO_3_) to a pH < 2, and 19 samples for the analysis of SO_4_^2−^ and Cl^−^ were preserved at a temperature below 4 °C, along with their respective field and transport. Water samples with turbidity values lower than 1 NTU were not filtered. The samples were transported to the Laboratory of the Instituto Nacional de Investigación en Glaciares y Ecosistemas de Montaña (INAIGEM) in Huaraz, where they were stored in a refrigerator before analysis.

Water samples were transported to the Chemistry Laboratory of the Instituto Geológico, Minero y Metalúrgico (INGEMMET) for analysis. Cl^−^ and SO_4_^2−^ were analyzed by ion chromatography (IC) using a Thermo Scientific Dionex, model ICS 5000+, manufacturer Thermo Fisher Scientific (Sunnyvale, CA, USA) [40], and Fe and S by inductively coupled plasma atomic emission spectrometry (ICP-AES) with Agilent Technologies Varian, model 730-ES, manufacturer Varian INC (Melbourne, Australia) [41], and Al, Cu, Co, Li, Mn, Ni, Pb and Zn by inductively coupled plasma mass spectrometry (ICP-MS) with Perkin Elmer, model NexION 300D, manufacturer Perkin Elmer (Shelton, CT, USA) [42]. Finally, the instruction protocols for the determination of ion parameters with IC and metals with ICP-AES and ICP-MS are specified in Appendix A.

#### 2.2.2. Quality Assurance and Quality Control (QA/QC)

All PFA and HDPE plastic containers were washed with diluted HNO_3_ for at least 24 h, then rinsed with deionized water and dried before use. The water used during sampling and analysis was ultrapure, and reagents were of high purity grade.

Transport and field blanks, as well as standards, were preserved and analyzed using the same procedures. On the other hand, two duplicate samples were randomly analyzed to verify that the analytical method provides precise results, aiming to achieve an acceptable relative percent difference (% RPD) between 0 and 20%.

The standards used were from Inorganic Ventures, traceable to the National Institute of Standards and Technology (NIST), and are specified in Appendix A. The standard deviations between duplicate samples were kept below 5%. The recovery rate varied between 80% and 112%, and the correlation coefficient for the curves of anions and metals was above 0.995.

### 2.3. Evaluation of Pollution and Human Health

#### 2.3.1. Environmental Quality Assessment 

The parameters studied were compared with the environmental quality standards for water established by Peruvian regulations (ECA-Peru) [43]. Two quality standards were considered, namely, (i) standards for waters that can be made potable with conventional treatment (ECA.1.A2), and (ii) standards for the surface waters of lagoons and rivers that are part of protected natural areas (ECA.4.E1 and ECA.4.E2).

#### 2.3.2. Heavy Metal Pollution Index

The Heavy Metal Pollution Index (HPI) is a tool that evaluates the overall relative load of water, considering multiple selected parameters. This index provides a comprehensive estimation of the impact of each individual metal on water quality, which allows for a comprehensive assessment of pollution. The HPI [44] was calculated using Equation (1) as follows:(1)$$\mathrm{HPI}=\frac{\sum_{i=1}^{n}\left(Q_{i}W_{i}\right)}{\sum_{i=1}^{n}\left(W_{i}\right)}$$
where n is the number of parameters considered, Q_i_ is the sub-index of the nth parameter, and W_i_ is the unit weight of the nth heavy metal parameter. The sub-index Q_i_ was calculated using Equation (2).
(2)$$Q_{i}=\frac{C_{i}}{S_{i}}\times 100$$

The value of C_i_ is the concentration in mg L^−1^ of the nth heavy metal parameter, and Si is the maximum permissible limit in drinking water set by the World Health Organization (WHO) [45] and the United States Environmental Protection Agency (US EPA) [46]. The proportionality constant (k) has a value of 1 [47,48,49]. The unit weight of the nth heavy metal parameter (W_i_) was calculated using Equation (3).
(3)$$W_{i}=\frac{k}{S_{i}}$$

The critical value of the HPI for drinking water is greater than 100 [44]. However, for a better categorization of pollution levels, the categorization proposed by Kawaya et al. [50], Kuraranidhi et al. [51], and Chakraborty et al. [52] was used, which is categorized in Appendix A. Furthermore, Geographic Information System (GIS) techniques were employed to spatially visualize the results of the HPI determination in the Negro River sub-basin. The Inverse Distance Weighting (IDW) method in ArcGIS 10.8 was used to interpolate the values from adjacent sampling sites [47,53,54]. The determination of HPI with parameters for Al and Mn is based on research by Tiwari et al. [54], Tengku et al. [55], and Razak et al. [56].

#### 2.3.3. Human Health Risk Assessment

Human exposure to metals in water primarily occurs through direct ingestion and dermal contact [53]. According to the United States Environmental Protection Agency (US EPA) [57,58], the amount of pollutants absorbed by the human body is calculated using the Chronic Daily Intake (CDI), which represents the amount of metal absorbed per kilogram of body weight per day, through ingestion, dermal absorption, or inhalation.

To calculate the CDI for ingestion and absorption, Equations (4) and (5) were used. CDI_in_ and CDI_d_ represent the exposure dose via ingestion and dermal absorption (µg kg^−1^ day^−1^), respectively, and C_i_ corresponds to the concentration of the nth metal in water (µg L^−1^). Table 1 details the values, units, and sources of the remaining parameters. Moreover, for those parameter concentrations below the analytical detection limit, half of the analytical detection limit was used [59].
(4)$${\mathrm{CDI}}_{d}=\frac{C_{i}\times IR\times {ABS}_{g}\times EF\times ED}{BW\times AT}$$
(5)$${\mathrm{CDI}}_{d}=\frac{C_{i}\times SA\times K_{p}\times {ABS}_{d}\times ET\times EF\times ED\times CF}{BW\times AT}$$

The Hazard Quotient (HQ) is estimated by comparing the CDI of each exposure route (direct ingestion and dermal absorption) with the reference dose (RfD) for each metal using Equation (6). The RfDin values for Fe, Li, Al, Co, Mn, Ni, and Zn are 700, 2, 1000, 0.3, 24, 20, and 300 µg/kg/day, and the RfDd values for Fe, Li, Al, Co, Mn, Ni, and Zn are 140, 10, 200, 0.06, 0.96, 0.8, and 75 µg/kg/day [62,63,64,65,66]. HQ values greater than 1 indicate that non-carcinogenic adverse effects should be considered for the specific exposure route [57,58].
(6)$$\mathrm{HQ}=\frac{\mathrm{CDI}}{\mathrm{RfD}}$$

Non-carcinogenic risk was determined using the Hazard Index (HI), which is the sum of the Hazard Quotients (HQs) caused by different exposure pathways for each metal, calculated using Equation (7). HQsin and HQsd are the hazard quotients resulting from direct ingestion and dermal absorption pathways, respectively. Similarly, if the HI exceeds 1, non-carcinogenic effects on human health should be considered [57,58].
(7)$$\mathrm{HI}=\sum \mathrm{HQ}={\;\mathrm{HQ}}_{\mathrm{in}}+{\mathrm{HQ}}_{d}$$

Additionally, to spatially visualize the HI results for each metal in the Negro River sub-basin, the Inverse Distance Weighting (IDW) method was employed in ArcGIS 10.8 [47,53,66].

### 2.4. Statistical Analysis

Various statistical analyses were conducted to identify the sources of contamination in the study. These analyses included Pearson correlation, Principal Component Analysis (PCA), and Cluster Analysis (CA) using Ward’s method with the Euclidean distance interval on standardized data. The suitability of the data was assessed using Bartlett’s test of sphericity (*p* < 0.001) and the Kaiser–Meyer–Olkin (KMO > 0.5) index. 

PCA is an effective tool for identifying common sources of contaminants. Unlike the Pearson correlation, which only measures individual linear relationships, PCA transforms the data set into principal components that reflect the most significant directions of variability. This allows for a clearer interpretation of how multiple pollutants interact and contribute to overall pollution, facilitating the identification of underlying patterns and possible sources of contamination [48,53].

Mathematical and statistical calculations were performed using Microsoft Excel 2020, R-Studio V 2022.02.0+443, and IBM SPSS 28.0. 

## 3. Results

### 3.1. Metal Concentration and Environmental Assessment

The in situ parameters of pH, DO, EC, TDS, salinity, temperature, ions (Cl^−^ and SO_4_^2−^), sulfur (S), and metals (Fe, Al, Cu, Co, Li, Mn, Pb, and Zn) are shown in Table 2. The pH values indicate the high acidity of surface water, high concentration of dissolved solids, and high electrical conductivity, which is evidenced by the high concentrations of metals reported. 

The average concentrations of the studied parameters decreased in the following order: SO_4_^2−^ (156.726 mg/L) > S (61.253 mg/L) > Fe (28.597 mg/L) > Al (3.832 mg/L) > Mn (1.085 mg/L) > Cl^−^ (0.637 mg/L) > Zn (0.234 mg/L) > Ni (0.085 mg/L) > Co (0.053 mg/L) > Li (0.036 mg/L) > Cu (0.005 mg/L) > Pb (0.002 mg/L). 

Furthermore, it was observed that, at certain sampling points, the concentrations of Cl^−^, Co, Li, Ni, and Pb were below the detection limit of the method used. These results are specifically detailed in Appendix A.

On the other hand, the results of the ion and metal analysis in the transportation and field blanks showed values below the method detection limit (MDL), indicating that potential cross-contaminations from processes and/or reagents are controlled. Additionally, the analysis of duplicates (QRure2-dup and RNegr-dup) showed a percentage of relative percent deviation (%RPD) within acceptable ranges.

The sampling points evaluated with the ECA-Peru are specified in Appendix A. The values of pH, Fe, and Mn at QRure2, QRure3, QUqui, RNegr, ROlle, and Al at QUqui (6.421 mg/L) were outside the range established by ECA.1.A2, indicating that the water is not suitable for human consumption. Similarly, the values of pH, EC, Zn, and Ni (except in LTara2) in LTara1, LTara2, and LTara3 were outside the range established by ECA.4.E1. On the other hand, the values of pH, EC, Zn, and Ni in QOtut, QPuma, QSNom3, QSNom5, and QSNom7 were outside the range established by ECA.4.E2; these concentrations affect the water quality of the Huascarán Park in the Negro River sub-basin. Additionally, significant concentrations of SO_4_^2−^, Fe, Al, Mn, Ni, Zn, Co, and Li were recorded in most surface water.

### 3.2. Evaluation of the Heavy Metal Pollution Index (HPI)

In this study, the HPI was used to describe the potential risk of Fe, Al, Cu, Mn, Ni, Pb, and Zn in the surface water of the Negro River sub-basin. The HPI results for each sampling point are shown in Appendix A. The average HPI value was 360.959, indicating high pollution. The HPI values ranged from 4.847 to 1627.805. Of these, 6 (32%) sampling points showed low pollution (HPI < 90), 2 (10%) had medium pollution (90 ≤ HPI < 150), and 11 (58%) exhibited high pollution (HPI ≥ 150). The unit weight (W_i_) of metals Pb, Ni, Al, and Fe was higher, indicating that a higher concentration of these metals in the water proportionally increases the HPI. Furthermore, Fe and Al substantially contributed to the HPI, which is consistent with their high concentration in water.

The spatial distribution of the HPI is observed in Figure 2. High contamination was evident at sampling points LTara1 (578.912), LTara2 (139.331), and LTara3 (268.704), representative of the waters of Tarahua lagoon and Rurec and Uruashraju snowcaps, as well as in QPuma (1627.805), for the Pumahuacanca snowcap, and in QSNom3 (853.902), for Huansan lagoon and Uruashraju snowcap with high contamination (HPI ≥ 150). These sampling points are representative of the origin of the water resource throughout the sub-basin.

Furthermore, it was observed that both medium and high contamination are distributed along the main course of the Rio Negro sub-basin, encompassing the streams Sin Nombre 3, 5, 7, Rurec, Pumahuaganga, Otuto, Uquian, and the rivers Negro and Olleros. Generally, with the exception of the sampling point QQuil (4.847), representative of the Quilloc stream; QArar (12.161), corresponding to the Araranca stream; QRure1 (12.331), representing the initial section of the Rurec stream; the sampling points QSNom2 (12.337) and QSNom4 (14.786), which represent the streams Sin Nombre 2 and 4; and the sampling point QPuyh (15.533) corresponding to the Puyhuan and Huaracayoc streams, the water is not suitable for human consumption. This affects the livelihoods developed in the sub-basin and may be due to the generation of ARD in the glacier and periglacial basin associated with the loss of glacier cover, specifically in the Rurec, Uruashrahu, Pumahuacanca, and Tuctopunta snow-capped peaks, and the geology of the area (Chicama Formation). 

### 3.3. Human Health Risk Assessment

The results of the Hazard Index (HI) values for the metals (Fe, Li, Al, Co, Mn, Ni, and Zn) are displayed in Figure 3. The HI for the studied metals ranged between 4.673 × 10^−5^ and 8.049 in children, and between 2.155 × 10^−5^ and 3.714 in adults. Moreover, the HI values in children and adults were in the following decreasing order: Co > Fe > Li > Mn > Al > Ni > Zn. On the other hand, the values of HQin were higher than HQd, indicating that the main route of non-carcinogenic risk was direct ingestion. It was also observed that the HI for children was higher than the HI for adults, indicating that children are more susceptible.

The results suggested that there is a non-carcinogenic risk (HI ≥ 1) to health caused by the metals Fe and Co in children and Fe, Li, and Co in adults. The HI values in children for Co were at the sampling points QRure2 (1.209), QRure3 (1.916), ROlle (2.129), RNegr (2.302), LTara3 (2.551), LTara1 (2.685), QPuma (2.760), QUqui (3.231), QSNom5 (5.356), QOtut (5.507), QSNom7 (6.489), and QSNom3 (8.049); for Fe, they were QPuma (3.064), QSNom3 (1.245), and QUqui (1.163), and Li was QSNom3 (2.003). Meanwhile, the HI values in adults for Co were RNegr (1.063), LTara3 (1.176), LTara1 (1.239), QPuma (1.274), QUqui (1.461), QSNom5 (2.472), QOtut (2.542), QSNom7 (2.995), and QSNom3 (3.715), and for Fe, the value was QPuma (1.414). On the other hand, the HI for Ni, Zn, Al, and Mn for all sampled points was less than 1, indicating no consideration of non-carcinogenic adverse effects in children and adults. Also, there are HI values close to 1, specifically for Fe in QSNom5 (0.691), QOtut (0.698), QSNom7 (0.844), and LTara1 (0.984) in children; QSNom5 (0.691) and QSNom3 (0.575) in adults; Li in QSNom3 (0.925) in adults; Mn in QSNom3 (0.543) in children, and Co in QRure2 (0.558), QRure3 (0.884), and ROlle (0.983) in adults.

The spatial distribution of HI ≥ 1 for Fe, Li, and Co is observed in Figure 4, and of HI < 1 in Appendix A. The spatial distribution of HI for all metals is higher in the upper basin, near the glacier and periglacial zone (Tarahua lagoon, Sin Nombre stream 3, and Pumahuaganga stream), and progressively decreases downstream to the Olleros River. The areas with HI ≥ 1 for Fe are the Pumahuaganga, Uquian streams, and Sin Nombre stream 3, and the area with HI ≥ 1 for Li is Sin Nombre stream 3. On the other hand, HI ≥ 1 for Co is distributed in all the main surface waters of the sub-basin, specifically in Tarahua lagoon, Otuto stream, Pumahuaganga, Rurec, Uquian, unnamed stream 3, 5, 7, and Negro and Olleros rivers.

### 3.4. Analysis of Pollutant Sources

The Pearson correlation, PCA, and CA are useful for identifying the possible origins of contaminants when they are interrelated. The parameters studied in the multivariate analysis were the pH, EC, TDS, salinity, SO_4_^2−^, Fe, S, Li, Al, Co, Mn, Ni, and Zn. The results of the Pearson correlation are shown in Figure 5. EC, TDS, salinity, SO_4_^2−^, Fe, S, Li, Al, Co, Mn, Ni, and Zn were positively correlated, and the correlation coefficients ranged between 0.367 and 0.999; on the other hand, the pH was negatively correlated with all the other parameters, with coefficients ranging from −0.896 to −0.385. All correlations were significant (*p* < 0.05) except between Li and the pH, SO_4_^2−^, and Fe (*p* > 0.05).

The feasibility results of the data, such as the Kaiser–Meyer–Olkin value (0.813) and Bartlett’s test of sphericity (*p* < 0.05), indicated that it was appropriate to perform a PCA. Two components with eigenvalues > 1 (PC1 with an eigenvalue of 11.036 and PC2 with 1.074) were extracted, explaining 93.155% of the total variance. The loading plot (Figure 6) showed that PC1 explained 84.896% of the total variance, with loadings for the pH (−0.854), EC (0.980), TDS (0.979), salinity (0.981), SO_4_^2−^ (0.949), Fe (0.808), S (0.966), Li (0.614), Al (0.967), Co (0.882), Mn (0.942), Ni (0.985), and Zn (0.994) being higher, and PC2 explained 8.259%, with the loading for Li (0.795) being higher.

The score plot (Figure 7) showed that QSNom3 had high scores on PC1 and PC2, indicating significant contamination with Li, Co, and Mn, and moderate contamination with Al and Ni; QPuma had a high score on PC1 and a low score on PC2, indicating considerable contamination with Fe, SO_4_^2−^, and S. On the other hand, sampling points QRure1, QQuil, QArar, QSNom2, QPuyh, QSNom4 had low scores on PC1, indicating a high pH and low contamination with Li, Co, Mn, Al, Ni, Zn, S, SO_4_^2−^, and Fe.

The CA dendrogram for sampling points is shown in Figure 8. Cluster 1 contained QQuil, QRure1, QArar, QSNom4, QPuyh, and QSNom2, representing water from lateral streams, which have a pH close to 7 and low metal concentration. Cluster 2 contained QPuma, QSNom3, QRure2, QRure3, LTara2, QOtut, QSNom5, QSNom7, RNegr, ROlle, LTara3, LTara1, and QUqui, representing surface water from the main channel, which has a pH less than 4 and high metal concentration. The CA dendrogram for parameters is shown in Figure 8. Cluster 1 contained EC, TDS, salinity, SO_4_^2−^, S, Fe, Co, Mn, Al, Ni, Zn, and Li, and cluster 2 contained the pH.

## 4. Discussion

### 4.1. Metal Concentration and Environmental Assessment

Environmental monitoring of the Olleros River, in 2013, during the dry season, reported the pH (3.51), Al (0.67 mg/L), Fe (4.374 mg/L), Mn (0.131 mg/L), and Ni (0.031); in 2014, the pH (4.14), Al (1.41 mg/L), Fe (3.833 mg/L), Mn (0.289 mg/L), and Ni (0.026 mg/L); and, in 2015, during the dry season, the pH (4.01), Al (2.247 mg/L), Fe (5.386 mg/L), Mn (0.523 mg/L), and Ni (0.040 mg/L), indicating a decrease in water quality over time [67,68,69]. This is influenced by rainfall; during the flood season, there is more precipitation, increasing flow and having a diluting effect. On the contrary, during the dry season, the concentration of metals increases and the pH lowers, which can dissolve the metal complexes in the sediment [30,31,50,65].

High concentrations of Fe, Al, Mn, Ni, Zn, Li, Co, SO_4_^2−^, S, and an acidic pH in surface water were manifested at the heads of glacier and periglacial basins, associated with factors such as lithology (Chicama Formation) and the loss of glacier cover, mainly for the surface water of Tarahua and Huansan lagoons, Pumahuaganga, Rurec, Otuto and Uquián streams, and Olleros and Negro rivers, indicating natural contamination [16,21,22,23,24,25,26,27,28]. The study conducted by Grande et al. [26] reported analogous values of the pH (4.405), EC (505.115 µS/cm), SO_4_^2−^ (224.231 mg/L), Al (4.007 mg/L), Fe (27.304 mg/L), Mn (0.946 mg/L), S (72.552 mg/L), Zn (0.261 mg/L), and Ni (0.075 mg/L) in surface water, coinciding that the contamination is natural, specifically by ARD, and highlighting that climatic, hydrological, geomorphological, geological, and biological factors can increase the generation of ARD. Likewise, Leyva [27] and Morales [28] reported a pH < 4 and a high concentration of Al, Fe, Mn, Cu, Ni, and SO_4_^2−^ in water, soil, and sediment in wetlands near Collotacocha lagoon and Uquián stream, highlighting the bioremediation capacity of wetland plant species. The main channel’s surface water is not suitable for human or animal consumption; furthermore, the metal concentration in waters representative of the sub-basin origin (LTara1, LTara2, QPuma, and QSNom3) decreases downstream due to the bioremediation of wetland flora, rainfall, and the precipitation of iron oxyhydroxysulfates [26,27,28,29].

Analogous studies in the Huascarán Park reported that surface water in the Quillcayhuanca and Shallap micro-basins had a pH < 4, highlighting that acidification is associated with lithology (Chicama Formation) [19]. Similarly, in the Quillcayhuanca micro-basin, a pH < 4 and high metal concentration (Al, Fe, Mn, Li, Ni, Co, and Zn) during dry and high-water seasons are of natural origin, associated with lithology (Chicama Formation). Likewise, the Pachacoto micro-basin showed a pH < 3 and high concentrations of Fe, Al, Mn, Zn, Co, Ni, and SO_4_^2−^ in springs and lagoons around the Pastoruri glacier due to the oxidation of pyrite-rich shales and sandstones (Chicama Formation) [17].

### 4.2. Evaluation of the Heavy Metal Pollution Index

The HPI values in the Negro River sub-basin were high; it was also evident that the main water channels exhibit high contamination. The categorization of the HPI is very variable [44,47,48,53,55]; therefore, the categorization proposed by Kawaya et al. [50], Karunanidhi et al. [51], and Chakraborty et al. [52] was used, as it better fit the water bodies contaminated with ARD. The Si values for Al, Zn, and Fe were not based on potential negative health effects but on organoleptic criteria [45,46]. Therefore, the HPI should be supported by other indices that evaluate comprehensive water quality.

### 4.3. Human Health Risk Assessment

It was demonstrated that the HI < 1 for Al, Ni, Mn, and Zn, indicating that adverse health effects need not be considered. However, adverse health effects (HI ≥ 1) caused by Fe, Li, and Co in children, and Fe and Co in adults should be considered. Additionally, the HI for certain sampling points was close to 1, mainly for Fe and Mn in children, and Fe, Li, and Co in adults. Cobalt induces neurotoxicity, tissue damage through free radicals, cardiovascular and endocrine dysfunctions, and inhibits RNA synthesis enzymes in humans after prolonged exposure [70,71,72]. High concentrations of Fe in the human body can cause oxidative stress, mitochondrial dysfunction, and neuroinflammation which may contribute to neurodegenerative disorders [73,74]. Li in drinking water has been associated with a decreased risk of suicide in patients with mood disorders; studies indicate it can affect the renal, endocrine, neurological, and gastrointestinal systems [75,76,77,78]. Mn is neurotoxic, and produces signs and symptoms similar to Parkinson’s disease, including the stimulation of the production of reactive oxygen species, neuroinflammation, disruption of neurotransmitter homeostasis, and neurodegeneration [73,79,80]. The uncertainty in the HI calculation has been mainly reported in the limitation of data and the RfD_in_ for Li and Co, so the p-RfD (provisional oral reference dose) was used, as well as the seasonal changes in metal concentration in water, and the daily water intake in the population of Olleros which cannot be quantified. Moreover, the parameters used were mostly from the US EPA, which might not apply to Peruvian populations due to interindividual variability [57,58]. Therefore, further research is needed to adapt and improve the risk assessment methodology for Peruvian populations; also, the risk assessment was individual, as the toxic mechanisms of each studied metal are different [81].

### 4.4. Analysis of Metal Contaminant Sources

There are various sources of metals, and each element can be derived from multiple different sources. The Pearson correlation, PCA, and CA classified the studied parameters into two distinct groups. The first group was defined by the pH, EC, TDS, salinity, SO_4_^2−^, Fe, S, Li, Al, Co, Mn, Ni, and Zn, and the second group was defined by Li; both groups were attributed to natural contamination from the geological source. The natural contamination of the water resource is supported by the accelerated loss of glacier cover, which exposes the underlying rock rich in sulfides (Chicama Formation) to oxidation and natural leaching in glacier and periglacial areas [16,21,22,23,24,25,26,82]. Furthermore, the probable presence of permafrost may increase the generation of ARD in periglacial zones [83,84]. The sampling points LTara1, LTara2, and LTara3, representative of surface waters from the Rurec and Uruashraju snowcaps, QPuma for the Uruashraju and Pumahuacanca snowcaps, and QSNom3 for Tuctopunta and Uruashraju are contaminated with ARD, with the Chicama geological formation composed of slate shales and metamorphic sandstones rich in sulfides. On the other hand, QRure1, which is representative of the Cashán snowcap, QSNom4 for Tuctu, QArar for Yanamarey, and QQuil for Pucaraju are not contaminated with ARD because the dominant geological formation is the Balotito of the Cordillera Blanca and, to a lesser extent, the Chicama Formation [16,26,27,28,82]. The water resource from the sampling points QSNom2 and QPuyh comes from the deglaciation of the Shacsha snowcap, dominated by the Balotito geology of the Cordillera Blanca, which is transported by the Oncor canal to the Huaracayoc and Puyhuan streams. The populated center of Olleros was not found contaminated with ARD [35,82]. Therefore, the generation of ARD is linked to deglaciation and the Chicama Formation.

The generation of ARD in glacial contexts is tied to the accelerated process of glacier melting and the geology of the area. Analogous studies in the European Alps, Pyrenees, and in Eagle Plains, Canada, showed high concentrations of metals and an acidic pH of geological origin, detailing that permafrost intensifies and concentrates the production of ARD [84,85,86,87,88]. Similarly, studies in the Quillcayhuanca, Shallap, and Pachacoto micro-basins showed contamination with ARD associated with the Chicama Formation and deglaciation [16,17,18,19,20]. Therefore, it is necessary to carry out geological and mineralogical studies to determine the rock’s capacity to generate ARD in periglacial and glacier environments; the same applies to studies in permafrost.

## 5. Conclusions

The high concentrations of the parameters Fe, Mn, Zn, Ni, Al, Co, Li, SO_4_^2−^, S, EC, pH, TDS, and the environmental assessment with the ECA-Peru, indicated that the surface water from the main channel is not suitable for human consumption and causes negative effects on the development of livelihoods in the Huascarán National Park. The average HPI was 360.959, indicating high pollution, with Fe, Al, and Mn contributing the most to the index. The spatial distribution was heterogeneous, indicating that the ecological risk is higher in the headwaters and decreases downstream, corresponding to the waters of the main channel (Rurec, Pumahuaganga, Uquian, and Otuto streams, Sin Nombre streams 7, 5, 3, Tarahua lagoon, and Negro and Olleros rivers).

The non-carcinogenic risk was assessed using the Hazard Index (HI), indicating that there is a non-carcinogenic risk caused by Fe, Li, and Co for children, and Fe and Co for adults. Furthermore, Fe and Al substantially contributed to the HPI, which aligns with their high concentration in water. The spatial distribution of the HI for all metals was higher in the upper basin. HI ≥ 1 for Fe was found in Pumahuaganga, Uquian streams, and Sin Nombre stream 3; HI ≥ 1 for Li was found in Sin Nombre stream 3, and HI ≥ 1 for Co was distributed throughout the sub-basin. The results of the correlation and multivariate analysis (PCA and CA) showed that the parameters pH, EC, TDS, salinity, SO_4_^2−^, Fe, S, Li, Al, Co, Mn, Ni, and Zn were attributed to geological sources which, due to deglaciation processes, expose the underlying rock to oxidation and leaching, generating ARD in the glacier and periglacial headwaters.

Health risk assessments stemming from water pollution with ARD are crucial for understanding the sources and levels of present contaminants. Exposure to metals such as Fe, Li, Al, Co, Mn, Ni, Zn can have severe consequences, ranging from acute poisonings to subchronic or chronic effects on inhabitants. This identification allows for an accurate assessment of the hazards faced by the exposed population, facilitating the implementation of appropriate preventative and corrective measures, and encouraging decontamination projects. Moreover, it provides valuable information for allocating resources efficiently, and strategically planning in public health. Knowing which metals are present in dangerous concentrations and in which specific areas allows efforts to be directed towards mitigating these risks and prioritizing interventions where the risk is highest. Ultimately, this thorough assessment is essential for protecting the health of communities and guiding informed decision-making in environmental health, providing a robust scientific basis for future health interventions and public health policies based on rigorous scientific data.

## Figures and Tables

**Figure 1 ijerph-21-00733-f001:**
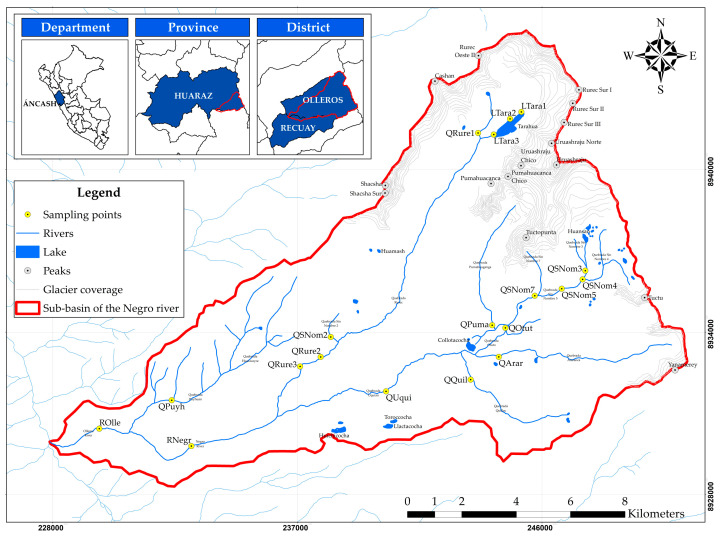
Locations of surface water sampling points in the Negro River sub-basin, Áncash, Peru.

**Figure 2 ijerph-21-00733-f002:**
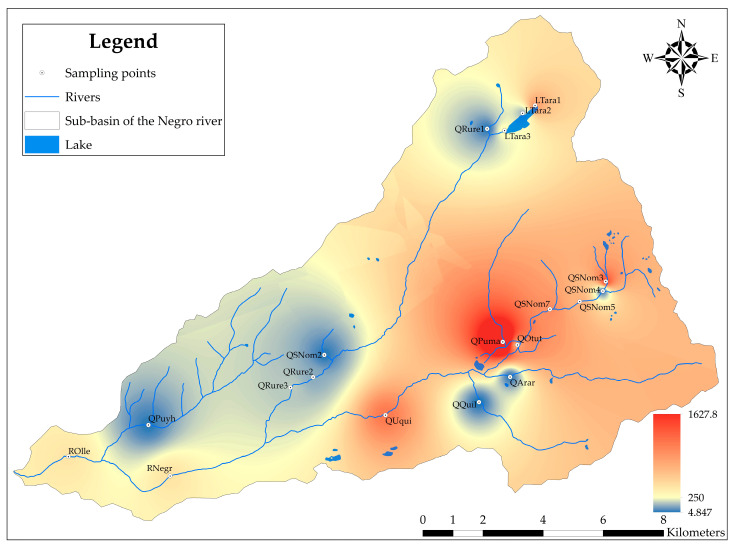
Geospatial distribution of the HPI in the Negro River sub-basin, Áncash, Peru.

**Figure 3 ijerph-21-00733-f003:**
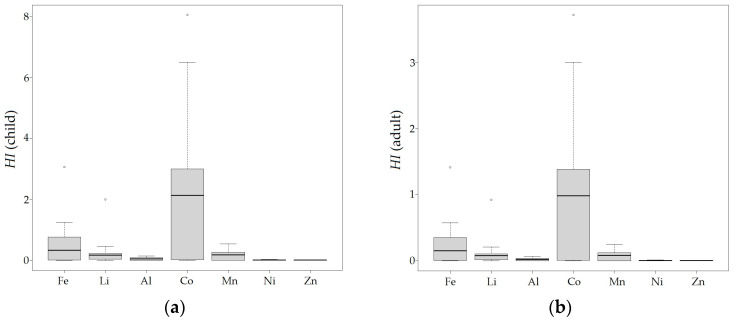
HI for iron (Fe), lithium (Li), aluminum (Al), cobalt (Co), manganese (Mn), nickel (Ni), and zinc (Zn). (**a**) HI for children; (**b**) HI for adults.

**Figure 4 ijerph-21-00733-f004:**
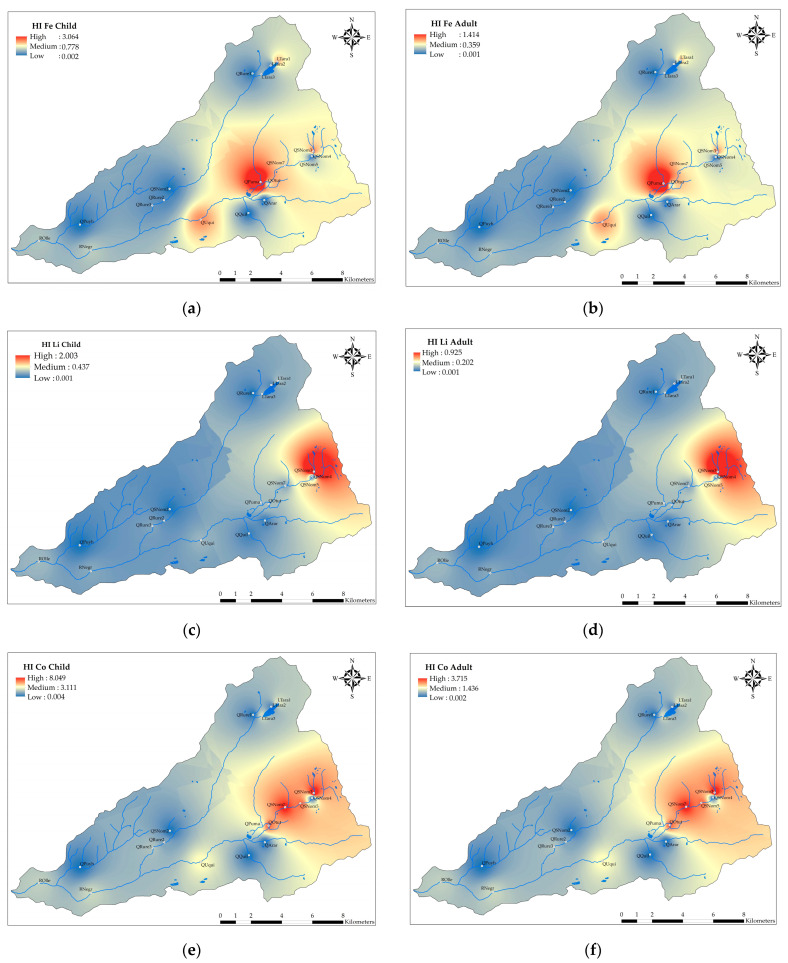
Spatial distribution of HI for Fe, Li, and Co: (**a**) HI for Fe in children; (**b**) HI for Fe in adults; (**c**) HI for Li in children; (**d**) HI for Li in adults; (**e**) HI for Co in children; and (**f**) HI for Co in adults.

**Figure 5 ijerph-21-00733-f005:**
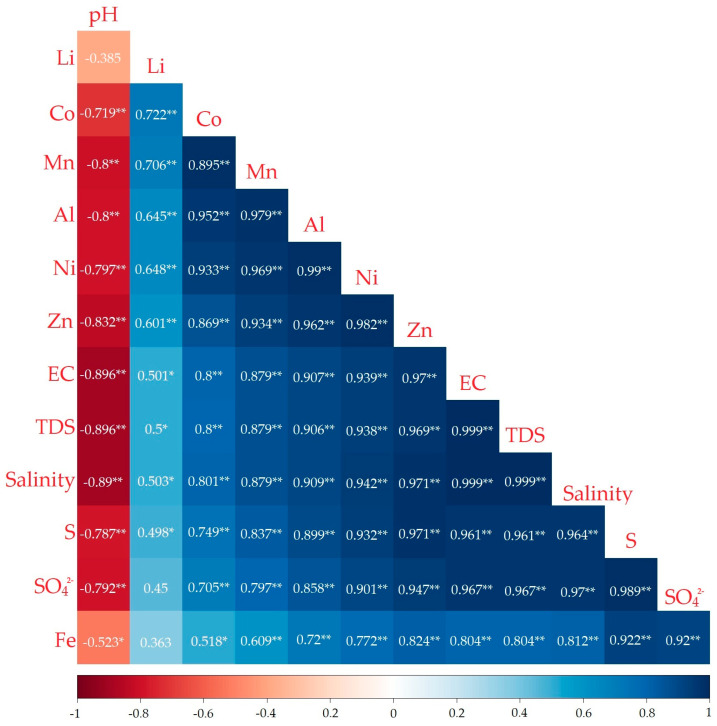
Pearson correlation coefficients for the parameters pH, EC, TDS, salinity, Li, Co, Mn, Al, Ni, Zn, Fe, S, and SO_4_^2−^ from surface water samples of the Río Negro sub-basin; (*) indicates significant correlation at *p* < 0.05 (two-tailed) and (**) indicates significant correlation at *p* < 0.01 (two-tailed).

**Figure 6 ijerph-21-00733-f006:**
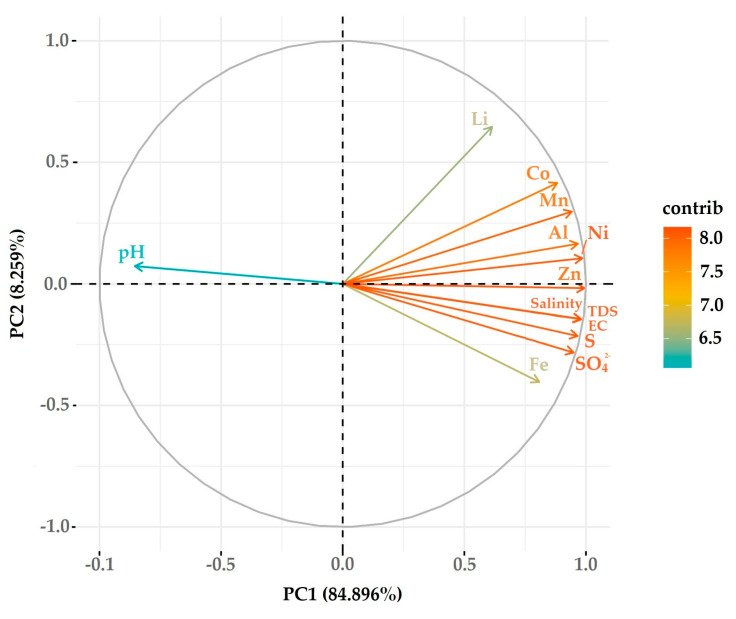
PCA loadings plot.

**Figure 7 ijerph-21-00733-f007:**
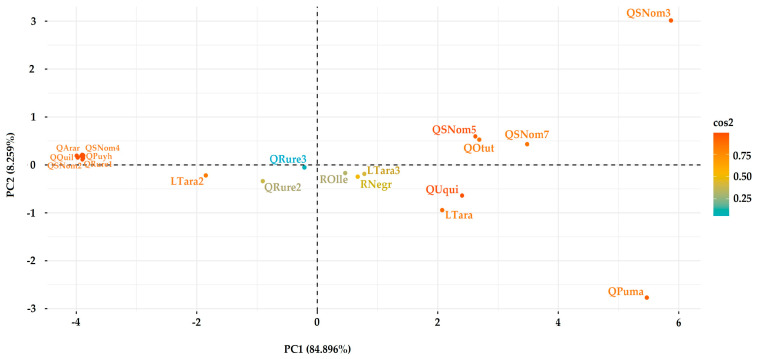
PCA scores plot.

**Figure 8 ijerph-21-00733-f008:**
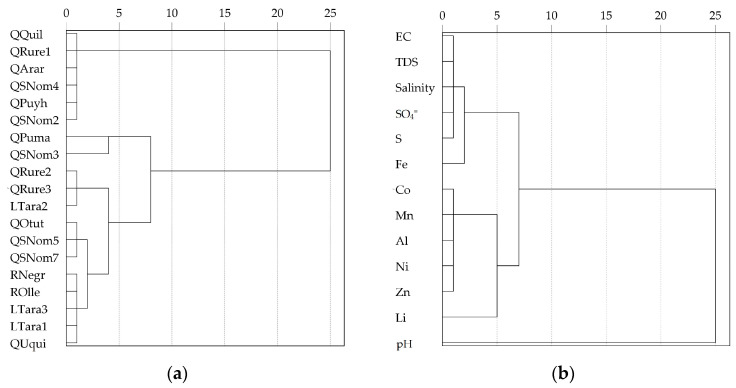
Dendrogram of sampling points and parameters. (**a**) Dendrogram by sampling points from the Negro River sub-basin; (**b**) dendrogram by the parameters studied in surface water from the Negro River sub-basin.

**Table 1 ijerph-21-00733-t001:** Detail of parameters for estimating Chronic Daily Intake (CDI).

Exposure Parameters	Unit	Reference	
		Children	Adults
Intake rate (IR) [59]	L day^−1^	1	2
Gastrointestinal absorption factor (ABS_g_) [48,58,60]	-	20% (Zn), 6% (Mn), 4% (Ni) and 20% (Fe, Li, Al and Co)	
Exposure frequency (EF) [59]	day year^−1^	365	
Exposure duration (ED) [59]	year	6	30
Average weight body weight (BW) [59]	kg	15	65
Average lifetime (AT) [59]	day	2094	1815
Skin-surface area (SA) [59]	cm^2^	8750	19,400
Permeability coefficient (K_p_) [58]	cm day^−1^	6 × 10^−4^ (Zn), 4 × 10^−4^ (Co), 1 × 10^−3^ (Fe, Li, Al, Mn and Ni)	
Dermal absorption factor (ABS_d_) [48,58]	-	0.1% (Fe, Li, Al, Co, Mn, Ni and Zn)	
Exposure time (ET) [61]	hour day^−1^	0.14	0.25
Conversion factor (CF) [47,61]	L cm^−3^	10^−3^	

**Table 2 ijerph-21-00733-t002:** Descriptive statistics of the studied parameters.

Parameters	Unit	Range	Mean	Median	SD	Coefficient of Variation
pH	unidad pH	2.708–6.912	4.077	2.998	1.71	41.945%
DO	mg/L	4.972–7.784	6.610	6.538	0.846	12.800%
EC	µS/cm	69.4–1848.6	826.653	997.6	589.037	71.256%
TDS	ppm	34.4–924.6	413.432	499	294.936	71.338%
Salinity	PSU	0.03–0.948	0.413	0.5	0.300	72.528%
Temperature	°C	1.268–14.35	9.260	9.682	3.688	39.823%
Turbidity	NTU	0.071–0.323	0.128	0.095	0.075	58.593%
Cl^−^	mg/L	0.1–1.8	0.637	0.6	0.512	80.503%
SO_4_^2−^	mg/L	1.6–493.8	156.726	167.3	133.647	85.274%
Fe	mg/L	0.110–160.85	28.597	17.54	38.863	135.897%
Al	mg/L	0.022–10.573	3.832	3.761	3.486	90.978%
Cu	mg/L	0.001–0.008	0.005	0.006	0.003	55.017%
Co	mg/L	0.000–0.181	0.053	0.0479	0.055	103.991%
Li	mg/L	0.000–0.305	0.036	0.0265	0.066	181.658%
Mn	mg/L	0.003–3.258	1.085	1.0997	0.980	90.230%
Ni	mg/L	0.000–0.237	0.085	0.088	0.078	91.704%
Pb	mg/L	0.000–0.004	0.002	0.0013	0.001	62.836%
S	mg/L	0.9–191.4	61.253	63.5	52.972	86.482%
Zn	mg/L	0.008–0.555	0.234	0.249	0.184	78.666%

## Data Availability

Researchers can request the data set from the Instituto Geológico, Minero y Metalúrgico with Test Report No. 140-2022-INGEMMET/DL-LQ.

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
