# Peer review of "Assessment of Human Health Risk Indices Due to Metal Contamination in the Surface Water of the Negro River Sub-Basin, Áncash"

_ijerph, 2024, doi:10.3390/ijerph21060733_

Round 1

Reviewer 1 Report

Comments and Suggestions for Authors

Thank you for the opportunity to review this manuscript. The manuscript “Assessment of human health risk indices due to metal contamination in surface water of the Negro river sub-basin, Áncash” presents an important topic. The study is well conducted and presented. I have few suggestions for further improvement as mentioned below:

1.     The introduction section is well written. However, background information about metal contamination is not sufficient. It will be good to understand the main theme of the paper (metals contamination) before moving to the analysis part.

2.     The authors have not provided reference values (about studied metals). For the sake of understanding and comparison, reference values suggested by world health organization (WHO), or any other watchdog will be useful. The values can be incorporated in text or in the tables will serve best.

3.     In the last paragraph (before conclusion section), the authors have very well provided information about previous relevant studies from different countries. However, studies from Asia region will be informative for comparison and support. A couple of studies from similar settings that I have pointed out (cited below) will be useful to include.

a.     Detection of arsenic (As), antimony (Sb) and bacterial contamination in drinking water

b.     Detection of heavy metals (Pb, Sb, Al, As) through atomic absorption spectroscopy from drinking water of District Pishin, Balochistan, Pakistan

4.     The analysis protocol used for the detection of studied parameters is limited. Further details there will be informative.

5.     The results of principal component analysis are useful and really informative. However, one would still think that correlation coefficients will suffice. It will be good to describe briefly the use of principal component analysis in this study.

Comments on the Quality of English Language

None

Author Response

We sincerely appreciate the time and effort you have dedicated to reviewing our manuscript. We highly value your detailed and constructive observations, which we have taken into account to make significant improvements to our work.

In the resubmitted document, you will find all the corrections and modifications clearly indicated, using the track changes feature to facilitate your review. I have also attached a detailed response to each of the points you mentioned, ensuring that all your concerns have been thoroughly addressed.

Reviewer 2 Report

Comments and Suggestions for Authors

1 I feel that the Introduction section is not comprehensive. There is need to provide sufficient background information so that readers can easily understand. The author's summary on the reduction of glacier cover area leading to changes in heavy metals in water bodies is insufficient. The research progress and some conclusions on the increase of metal concentration in water bodies caused by the loss of glacier cover should be concluded.

2 Quantification limits should be supplied about inductively coupled plasma atomic emission spectrometry (ICP-AES).

3 Line200-line 205: The expression is unclear. What is the proportion of undetected parameters? How to include in mean statistics.

4 Line 216 Why did the Co and Li element not be calculated when only some elements were selected in the HPI calculation? Please explain.

5 Figure 5. The metal elements used in the image and annotations are inconsistent. Please explain the reason. Only 6 metal elements in the picture were included in the correlation statistics? Why not discuss other metal elements.

6 Conclusion part should be rewritten to show what is the significance of your work for the study and to go beyond the results sections for forming the conclusions.

Author Response

We are immensely grateful for the time and attention you have devoted to reviewing our manuscript. Your insightful and detailed feedback has been invaluable, guiding us to enhance the quality and coherence of our work significantly.

In response to your comments, we have meticulously revised the manuscript and resubmitted it with all changes clearly marked using the track changes feature. This is to ensure ease of review and to transparently show how your suggestions have been implemented.

Reviewer 3 Report

Comments and Suggestions for Authors

In this study, the authors evaluated human health risk indices due to metal contamination in surface water of the Negro River sub-basin, Áncash, Peru, which is caused by acid rock drainage (ARD) generation in glacial and periglacial environments. They reported that adverse health effects induced by Fe, Li and Co in children and adults should be considered. The reviewer thinks that the results presented here are valuable from a social point of view, because similar studies are still scarce. My comments are given below.

 Significant figures of values: The significant figures of values such as concentration, HPI and HI presented in this paper are too many. They are at most three based on analytical errors and uncertainty in hypotheses.

Lines 26 and 66: The authors should use SO42- instead of SO4=. Moreover, it is strange to discuss SO42- and S separately, because major S species in water is SO42-.. Confirm this by calculating the ratio of sulfate-S to total S measured by ICP-AES.

Lines 68–74: The reviewer thinks that the human health risks directly associated with acidic water itself should be also evaluated. What do the authors think about this?

Lines 97–103: The authors did not measure suspended solid (SS) in the water samples. The reviewer thinks that the water samples contain significant amounts of SS, which may affect strongly the health risk of heavy metals dependent on their forms, i.e., dissolved or particulate form in the water.  

Lines 104–105: Were the acidified samples filtrated prior to ICP analysis?

Lines 119–120: The result of these blanks should be shown.

Line 122: Provide the manufacturer and product number of standards used.

Line 133: Correct “3.3.2” to “2.3.2”. Moreover, write the heading in English.

Line 155: Correct “3.3.3” to “2.3.3”.

Lines 168–174: Explain the reason why Cu and Pb are not eligible for estimating their HQ.

Author Response

We deeply appreciate the time and thoughtful effort you have dedicated to reviewing our manuscript. Your detailed feedback has been instrumental in helping us improve both the clarity and depth of our work.

In the revised document that we have resubmitted, you will find all modifications and corrections clearly highlighted using the track changes feature. This is intended to facilitate your review process and ensure transparency. Additionally, attached is a comprehensive response to each of the points you raised, confirming that we have addressed all of your concerns thoroughly.

Round 2

Reviewer 2 Report

Comments and Suggestions for Authors

I found the authors have satisfactorily responded to all my comments with a level of care. Accordingly, I hereby recommend acceptance without further revision.